# Spirulina Supplements as a Source of Mineral Nutrients in the Daily Diet

Katarzyna Janda-Milczarek [1,*], Kinga Szymczykowska [1], Karolina Jakubczyk [1], Patrycja Kupnicka [2], Karolina Skonieczna-Żydecka [3], Bogumiła Pilarczyk [4], Agnieszka Tomza-Marciniak [4], Alicja Ligenza [1], Ewa Stachowska [1] and Bartosz Dalewski [5]

[1] Department of Human Nutrition and Metabolomics, Pomeranian Medical University in Szczecin, 24 Broniewskiego Street, 71-460 Szczecin, Poland

[2] Department of Biochemistry and Medical Chemistry, Pomeranian Medical University in Szczecin, 72 Powstańców Wlkp. Street, 70-111 Szczecin, Poland

[3] Department of Biochemical Sciences, Pomeranian Medical University in Szczecin, 24 Broniewskiego Street, 71-460 Szczecin, Poland

[4] Department of Animal Reproduction Biotechnology and Environmental Hygiene, Faculty of Biotechnology and Animal Husbandry, West Pomeranian University of Technology, 71-270 Szczecin, Poland

[5] Department of Dental Prosthetics, Pomeranian Medical University in Szczecin, Powstańców Wlkp. 71 Street, 70-111 Szczecin, Poland

* Correspondence: katarzyna.janda.milczarek@pum.edu.pl

**Abstract:** Spirulina is a microalga cultivated in many countries. It is a source of valuable protein, polyunsaturated fatty acids, vitamins, antioxidants and elements. We have not found studies that address the effect of supplement form or cultivation method on the mineral content of spirulina supplements. The aim of this study was to determine whether supplement form (tablet and powder) and cultivation method (organic and conventional) of spirulina have a bearing on the mineral nutrients content. Such an approach accounts for the innovation of our research. The material used in the study was spirulina in tablets and powder form, marketed as a dietary supplement. Samples were analyzed using inductively coupled plasma optical emission spectrometry (ICP-OES). In turn, selenium (Se) content was determined by spectrofluorimetry. Overall, in terms of mean values, the most abundant mineral in spirulina supplements was phosphorus (P) (15,149 mg/kg) and the least abundant was Se (0.31 mg/kg). Our findings show that both supplement form and cultivation method affect the mineral content of spirulina. Supplements in powder form had a significantly higher content of important elements, such as iron (Fe) (673 mg/kg), magnesium (Mg) (4151 mg/kg) and potassium (K) (16,686 mg/kg), while at the same time containing significantly less sodium (Na) (9868 mg/kg). In terms of the cultivation method, organic spirulina supplements turned out to be a richer dietary source of Fe (703 mg/kg) and K (14,893 mg/kg). In turn, conventionally grown supplements had higher contents of calcium (Ca) (11,269 mg/kg), phosphorus (P) (16,314 mg/kg) and strontium (Sr) (47 mg/kg). Spirulina can therefore be a valuable addition to the daily diet, helping people to achieve the required intake of micronutrients.

**Keywords:** *Arthrospira*; conventional growing; elements; organic growing pills; powdery





## 1. Introduction

Spirulina (*Arthrospira* sp.) is a microalga that has been consumed for centuries by indigenous peoples in Mexico and northern Africa [1]. The alkaline lakes which can be found in those regions are its natural habitat [2]. Spirulina belongs to the phylum cyanobacteria, and the term covers two species of microalgae—*Arthrospira platensis* and *Arthrospira maxima*. Both are used in the production of food and dietary supplements [3,4]. Spirulina is capable of oxygenic photosynthesis and reproduces asexually by binary fission until it

reaches its mature status [5]. Spirulina is currently the most popular commercially grown algae. The main producers are the DIC group of companies, Earthrise in California (USA) and Hainan DIC Marketing on Hainan Island (China), as well as Cyanotech Corporation of Hawaii. Smaller producers are located primarily in the Asia-Pacific region (mainly in China and India) [6], although there are plans to grow this alga in Europe as well [7]. Briefly, the harvesting of spirulina occurs from natural habitats that are difficult to control in view of the chemical composition of the environment, and from environments where the environmental parameters (temperature, pH, light, chemical composition of the culture medium, among others) are under human control [6,8]. The protein content of *Arthrospira* can be as high as 70%, and it is protein of high biological value, due to its content of all essential amino acids in the proportions recommended by the FAO (Food and Agriculture Organization). Consequently, it is superior to other plant sources of protein and as valuable as eggs, meat and milk [9]. Of particular note is the content of polyunsaturated fatty acids (PUFAs), such as eicosapentaenoic acid (EPA) and docosahexaenoic acid (DHA) [10]. It is also important to mention the vitamin content—A, D, E, K and group B vitamins. The high content of vitamin B12 makes spirulina particularly valuable for vegans and vegetarians who, due to the lack of meat products in their diet, may be at risk of deficiencies of this vitamin [11]. Spirulina is a good source of mineral nutrients such as Ca, Fe, Se, F or I [12–15]. It also contains high amounts of carotenoids (astaxanthin, zeaxanthin, β-carotene), polyphenols and chlorophyll [10,11,16]. With such a plethora of bioactive compounds, *Arthrospira* boasts antioxidant, anti-inflammatory and immunomodulatory properties [17]. It has also been found to inhibit the activity of viruses, including HIV and influenza virus [18]. The antioxidant effects of spirulina can be attributed to its ability to regulate antioxidant enzymes, such as superoxide dismutase, catalase and glutathione peroxidase. Spirulina also inhibits cyclooxygenase-2, as well as inducible nitric oxide synthase gene expression. Findings from animal studies suggest its protective effect against the harmful effects of heavy metals [17]. Spirulina has been shown to effectively lower triacylglycerols and LDL cholesterol in the blood and to help lower blood pressure [19]. In addition, it has been reported to have a blood-glucose-lowering effect in diabetes patients [16]. Its effects also include aiding weight loss in obese people and improving metabolic parameters, e.g., increasing insulin sensitivity, reducing inflammation, lowering total and LDL cholesterol and increasing HDL cholesterol [20]. Preliminary clinical studies have suggested that spirulina may exert neuroprotective effects, particularly in malnourished children, by supporting brain development and improving language and motor skills. In addition, it can be beneficial in alleviating mental and physical fatigue [21]. In in vitro research, it has been demonstrated that spirulina has a positive effect on the abundance of the human microbiota, which means that it could be used as a prebiotic [22]. Despite numerous studies conducted in vitro and in animal models, the effects of spirulina on the human body have not been fully investigated and confirmed. Algae also play a major role in other sectors, such as being used to produce biofuels [23,24], biopolymers, feedstocks and fertilizers. This contributes to increased interest in this sector by European Union structures [7]. The form of the supplement can presumably affect its quality, both the activity of the biologically active compounds and the elemental content, but no literature was found on this subject. Some authors have studied spirulina supplements in powder and tablet form. However, in the analysis of the results, they do not take into account the effect of the form of the supplement on the content of the studied parameters, including the content of elements [25]. While there are studies confirming that spirulina is rich in numerous minerals and other health-promoting compounds [12–15,17,26], we have not found studies that would address the effect of supplement form or cultivation method on the mineral content of spirulina supplements—this approach makes our study innovative.

The aim of this study was to determine the mineral nutrients content in spirulina supplements. We also investigated whether supplement form and cultivation method of spirulina have a bearing on the mineral nutrients content.

## 2. Materials and Methods

### 2.1. Material

The material used in the study was spirulina in tablet and powder form, marketed as a dietary supplement. A total of 33 spirulina samples from different countries of origin were obtained from specialist shops: 26 supplements were in tablet form, 2 in powder form and 5 in capsule form (Table 1). Table 1 shows the country where the supplement company is headquartered. Determining the origin of the spirulina proved to be a problem, as in only two cases did the label on the box indicate where the spirulina came from. Therefore, this information was not included in the table. For the purpose of analysis, the capsules were opened, their contents poured out and the sample was treated as a powder. Therefore, a total of 7 supplements were in powder form. The samples in tablet form were ground thoroughly using a mortar and pestle under aseptic conditions before the analysis. The samples were stored at room temperature until analysis.

**Table 1.** Description of tested samples of spirulina supplements.

| No. | Supplier | Company Registered Office | Supplement Form | Cultivation Method |
|---|---|---|---|---|
| 1 | NATURE'S WAY | USA | powder (capsules) | conventional |
| 2 | SWANSON | USA | powder (capsules) | conventional |
| 3 | SOLARAY | USA | powder (capsules) | conventional |
| 4 | SPRING VALLEY | USA | powder (capsules) | conventional |
| 5 | MYVITA | Poland | tablets | conventional |
| 6 | TRZY ZIARNA | Poland | tablets | conventional |
| 7 | PROSTO Z NATURY WITPAK | Poland | tablets | conventional |
| 8 | PLENUS | Poland | tablets | conventional |
| 9 | AURA HERBALS | Poland | tablets | conventional |
| 10 | NATUR PLANET | Poland | powder | conventional |
| 11 | GAL | Poland | powder (capsules) | conventional |
| 12 | AGNEX | Poland | tablets | conventional |
| 13 | VEGANICITY (HEALTH PLUS) | UK | tablets | conventional |
| 14 | HAYA LABS | Russia | tablets | conventional |
| 15 | SOLGAR | USA | tablets | conventional |
| 16 | PURITAN'S PRIDE | USA | tablets | conventional |
| 17 | STAR | Czech Republic | tablets | conventional |
| 18 | THE VITAMIN SHOPPE | USA | tablets | conventional |
| 19 | SWANSON | USA | tablets | conventional |
| 20 | AGNEX | Poland | powder | conventional |
| 21 | KENAY | Poland | tablets | conventional |
| 22 | ALG-BORJE | Island | tablets | conventional |
| 23 | HANOJU | Germany | tablets | conventional |
| 24 | VITATREND | Germany | tablets | organic |
| 25 | BIO ORGANIC FOODS | Poland | tablets | organic |
| 26 | DONUM NATUREA | European Union | tablets | organic |
| 27 | MYVITA | Poland | tablets | organic |
| 28 | NOW | USA | tablets | organic |
| 29 | HANOJU | Germany | tablets | organic |
| 30 | PUKKA | UK | tablets | organic |
| 31 | RAINFOREST FOODS | UK | tablets | organic |
| 32 | NATURES'S MULTI-VITAMIN (PURE HAWAIIAN) | USA | tablets | organic |
| 33 | VENUSTI | Poland | tablets | organic |

### 2.2. Sample Preparation

Samples were mineralized using a microwave mineralization system MARS 5, CEM (CEM Corporation Matthews, NC 28106, USA). Briefly, samples (0.8 mL) were transferred to polypropylene tubes. To the vials, 2 mL of 65% $HNO_3$ (Suprapur, Merck, Darmstadt, Germany) was added and allowed to pre-react for 30 min. Then, 0.5 mL of unstabilized 30% $H_2O_2$ solution (Suprapur, Merck, Darmstadt, Germany) was added to the vials. After

all reagents were added, samples were placed in Teflon dishes and heated (35 min, 180 °C). After that, the samples were cooled to room temperature. In a clean hood, the samples were transferred to acid-washed 15 mL tubes. Samples were diluted 5 times before measurement. Samples (2 mL) were enriched with internal standard to obtain a final concentration of 0.5 mg/L yttrium, 1 mL of 1% Triton (Triton X-100, Sigma Kawasaki, Japan) and diluted to a final volume of 10 mL with 0.075% nitric acid (Suprapur, Merck, Darmstadt, Germany). Blank samples were prepared by adding concentrated nitric acid (500 μL) to the tubes and then diluted in the same method as the test samples. Multi-element calibration standards (ICP multi-element standard solution IV, Merck, Darmstadt, Germany) were prepared with different concentrations of inorganic elements. Deionized water (Direct Q UV, Millipore, approx. 18.0 MΩ) was used to prepare the solutions.

*2.3. Sample Determination*

Briefly, samples were analyzed using inductively coupled plasma optical emission spectrometry (ICP-OES, ICAP 7400 Duo, Thermo Scientific, Waltham, MA USA), which allows measurement of various elements, including in plant samples [27,28]. An ICP-OES with a concentric nebulizer and cyclonic spray chamber was used. The analysis was performed in radial and axial modes. The wavelengths used in the analysis were Zn 206,200, Cr 205,560, Mn 257,610, Cu 224,700, Fe 259,940. Validation was performed by evaluating reference material NIST SRM 8414 (National Institute of Standards and Technology, Gaithersburg, MA, USA), limits of detection (LOD) and recovery of internal standard (yttrium) (Table 2). To eliminate potential interferences, emission lines were empirically selected in a pilot measurement. Such a validation model is often used in ICP-OES studies [28]. Y recovery ranged from 90–106%. R2 values for all standard curves ranged from 0.998 to 1.000.

**Table 2.** Analysis of reference material Bovine Muscle NIST-SRM 8414, limits of detection (LOD), and relative sample deviation (% RSD) range [29].

| Element | Certified [mg/L] | Measured [mg/L] ($n$ = 3) | LOD [mg/L] | % RSD Range |
|---|---|---|---|---|
| Ca | 145 ± 20 | 141 | 0.00676 | 0.4–2.5 |
| Mn | 0.37 ± 0.09 | 0.43 | 0.00026 | 1.0–6.2 |
| K | 15,170 ± 370 | 15290 | 0.08426 | 0.3–1.4 |
| Zn | 142 ± 14 | 138 | 0.00065 | 1.5–5.7 |
| Cu | 2.84 ± 0.45 | 3.06 | 0.00186 | 2.4–8.2 |
| Fe | 71.2 ± 9.2 | 76.1 | 0.00022 | 1.8–6.7 |
| Na | 2100 ± 80 | 2146 | 0.08137 | 0.8–4.1 |
| Pb | 0.38 ± 0.24 | 0.48 | 0.00178 | 5.3–11.2 |
| Cr | 0.071 ± 0.038 | 0.080 | 0.00044 | 3.9–9.2 |
| P | 8360 ± 450 | 8874 | 0.00532 | 0.8–3.2 |
| Mg | 960 ± 95 | 923 | 0.00159 | 0.5–1.9 |

Briefly, selenium (Se) concentration was determined by spectrofluorimetry using a SHIMADZU RF-5001 PC analyzer. Samples were wet-digested in concentrated $HNO_3$ (230 °C/180 min) and $HClO_4$ (310 °C/20 min). Then, 9% HCl was added to reduce selenate VI to selenate IV. Subsequently, selenate IV was complexed with 2,3-diaminonaphthalene (Sigma), and the resulting complex was extracted with cyclohexane (Chempur, Piekary Śląskie, Poland). Fluorescence was measured at an emission wavelength of 518 nm and an excitation wavelength of 378 nm. The accuracy of the method was based on NCS-ZC 71001 reference material (certified reference material—BCR-185R bovine liver) from China NatiAnalysis Center for Iron and Steel (Beijing, China). The determined Se concentration was 90.9% of the standard value. Two replicates were performed for each sample, and the average of the data, expressed in milligrams per kilogram wet weight, was used in the statistical analysis. The certified selenium concentration was 1.68 μg/g, the labeled concentration was 1.56 μg/g, and the recovery was 93%.

Selenium (Se) concentration was determined by spectrofluorimetry with a SHIMADZU RF-5001 PC analyzer. Samples of spiruline were wet-digested in concentrated $HNO_3$ (230 °C/180 min) and $HClO_4$ (310 °C/20 min). An amount of 9% HCl was added to the digested samples to reduce selenate VI to selenate IV. Subsequently, selenate IV was complexed with 2,3-diaminonaftalene (Sigma) and the resulting complex was extracted with cyclohexane (Chempur). Fluorescence was measured from the organic layer (cyclohexane) at 518 nm emission wavelength and 378 nm excitation wavelength. The accuracy of the analytical method was based on NCS-ZC 71001 (beef liver) reference material from China NatiAnalysis Center for Iron and Steel (Beijing, China). The determined Se concentration was 90.9% of the standard value. Two replicates were performed for each sample, and statistical analysis used the average of the data, expressed in milligrams per kilogram wet weight. The reference material was Certified Reference Material BCR–185R (bovine liver). The certified selenium concentration was 1.68 µg/g, concentration determined—1.56 µg/g, recovery was 93%.

### 2.4. Statistical Analysis

All determinations were carried out in at least three replicates.

The obtained test results were examined using statistic tools to show the differences between observed groups (STATISTICA 13.0; StatSoft Inc., Palo Alto, CA, USA). Results were expressed as mean values and standard deviation; however, statistical significance of differences was determined based on median, upper quartile and lower quartile. Since the obtained data violated normality and demonstrated heterogeneous variability, non-parametric tests were used. The significance of differences between the bodybuilder and control group outcomes was assessed using the Mann–Whitney test. Differences were considered significant at $p \leq 0.05$.

### 3. Results

Overall, in terms of mean values, the most abundant mineral in spirulina supplements was phosphorus (P) (15,149 $\pm$ 13,024 mg/kg) and the least abundant was selenium (Se) (0.31 $\pm$ 0.91 mg/kg). The content of calcium (Ca) was 8554 $\pm$ 17,869 mg/kg, iron (Fe)—664 $\pm$ 436 mg/kg, magnesium (Mg)—3726 $\pm$ 1283 mg/kg, potassium (K)—14,274 $\pm$ 3628 mg/kg, sodium (Na)—13,439 $\pm$ 6793 mg/kg, and strontium (Sr) 42 $\pm$ 45 mg/kg. The high standard deviations were due to the large differences in mineral contents determined in individual samples (Table 3).

**Table 3.** Element content of tested samples ($n$ = 33) of spirulina supplements—results of statistical analysis.

| Elements | Mean | Standard Deviation | Min. | Max. |
|:---:|:---:|:---:|:---:|:---:|
| Ca | 8554.46 | 17,869.90 | 761.191 | 76,781.53 |
| Fe | 664.80 | 436.81 | 212.228 | 2785.82 |
| Mg | 3726.32 | 1283.81 | 2134.017 | 11,067.04 |
| K | 14,274.26 | 3628.57 | 9019.566 | 38,335.28 |
| Na | 13,439.40 | 6793.01 | 3102.046 | 36,144.06 |
| Sr | 42.83 | 45.59 | 6.784 | 258.42 |
| Zn | 45.96 | 71.57 | 7.068 | 322.14 |
| P | 15,149.59 | 13,024.69 | 7196.181 | 70,759.49 |
| Se | 0.31 | 0.91 | 0.031 | 5.69 |

Table 4 shows the content of the studied minerals depending on the form of the supplement. Supplements in powder form had statistically significantly higher content of Fe, Mg and K ($p$ = 0.042945, $p$ = 0.010622, $p$ = 0.001389, respectively). Tablets, on the other hand, contained significantly more Na ($p$ = 0.008072). With respect to the content of the other elements (Ca, Sr, Zn, P, Se), the differences in their content between tablets and powder were not statistically significant ($p$ > 0.05).

**Table 4.** Element content in spirulina supplements depending on the form of the supplement.

| Elements | Powder (*n* = 7) | | Tablets (*n* = 26) | | *p* Value |
|---|---|---|---|---|---|
| | **Mean** | **Standard Deviation** | **Mean** | **Standard Deviation** | |
| | Elements Content [mg/kg] | | | | |
| Ca | 4278.25 | 3256.790 | 9705.75 | 19,933.38 | 0.267682 |
| P | 11,225.16 | 2069.442 | 16,206.17 | 14,472.91 | 0.120306 |
| Mg | 4151.03 | 1687.599 | 3611.98 | 1138.20 | 0.010622 * |
| K | 16,686.65 | 5245.867 | 13,624.77 | 2755.92 | 0.001389 * |
| Na | 9868.43 | 5866.450 | 14,400.81 | 6736.23 | 0.008072 * |
| Fe | 673.10 | 205.513 | 662.57 | 481.51 | 0.042945 * |
| Zn | 22.73 | 12.172 | 52.21 | 79.34 | 0.081545 |
| Se | 0.13 | 0.048 | 0.36 | 1.02 | 0.700103 |
| Sr | 48.73 | 35.758 | 41.24 | 47.97 | 0.099426 |

* $p < 0.05$—statistically significant difference.

The mineral content of the supplements depending on the cultivation method of spirulina is shown in Table 5. Supplements containing organically grown spirulina had statistically significantly higher content of Fe, P and K ($p = 0.03088$, $p = 0.028598$, $p = 0.032089$, respectively) compared to the products of conventional cultivation. In turn, conventionally grown supplements had significantly higher contents of Ca and Sr ($p = 0.032089$ and $p = 0.033331$, respectively). In the case of the other elements (Mg, Na, Zn, Se), the differences were not statistically significant ($p > 0.05$).

**Table 5.** Element content of spirulina supplements depending on cultivation method.

| Elements | Conventional Growing (*n* = 23) | | Organic Growing (*n* = 10) | | *p* Value |
|---|---|---|---|---|---|
| | **Mean** | **Standard Deviation** | **Mean** | **Standard Deviation** | |
| | Elements Content [mg/kg] | | | | |
| Ca | 11,269.39 | 20,841.41 | 2310.14 | 1656.474 | 0.032089 * |
| P | 16,314.46 | 15,474.80 | 12,470.40 | 1051.740 | 0.028598 * |
| Mg | 3643.43 | 1309.15 | 3916.98 | 1223.508 | 0.160094 |
| K | 14,004.90 | 4138.59 | 14,893.80 | 1939.565 | 0.032089 * |
| Na | 13,610.53 | 7048.95 | 13,045.78 | 6260.908 | 0.924177 |
| Fe | 647.91 | 480.29 | 703.66 | 318.812 | 0.03088 * |
| Zn | 34.35 | 49.61 | 72.66 | 102.381 | 0.076691 |
| Se | 0.14 | 0.10 | 0.70 | 1.600 | 0.831172 |
| Sr | 47.00 | 52.01 | 33.23 | 23.307 | 0.033331 * |

* $p < 0.05$—statistically significant difference.

Seeing as organic spirulina supplements were only available in tablet form, their mineral content was checked for differences against tablets containing non-organic spirulina—the results are shown in Table 6.

**Table 6.** Element content of spirulina supplements in tablets depending on the of cultivation method.

| Elements | Conventional Growing (*n* = 16) | | Organic Growing (*n* = 10) | | *p* Value |
|---|---|---|---|---|---|
| | **Mean** | **Standard Deviation** | **Mean** | **Standard Deviation** | |
| | Elements Content [mg/kg] | | | | |
| Ca | 14,328.00 | 24,342.07 | 2310.14 | 1656.474 | 0.073099 |
| P | 18,541.02 | 18,111.00 | 12,470.40 | 1051.740 | 0.058103 |
| Mg | 3421.35 | 1049.88 | 3916.98 | 1223.508 | 0.007932 * |
| K | 12,831.63 | 2907.13 | 14,893.80 | 1939.565 | 0.000419 * |
| Na | 15,247.70 | 6946.20 | 13,045.78 | 6260.908 | 0.450316 |
| Fe | 636.89 | 561.58 | 703.66 | 318.812 | 0.001533 * |
| Zn | 39.43 | 58.40 | 72.66 | 102.381 | 0.215863 |
| Se | 0.15 | 0.11 | 0.70 | 1.600 | 0.527603 |
| Sr | 46.24 | 58.03 | 33.23 | 23.307 | 0.103549 |

* $p < 0.05$—statistically significant difference.

Statistical analysis of the results revealed significant differences in the content of Mg, K and Fe ($p = 0.007932$, $p = 0.000419$ and $p = 0.001533$, respectively), with organic spirulina tablets being the richer source.

Detailed results, including minimums, maximums, medians and quartiles are presented as a Supplement (Tables S1–S4).

## 4. Discussion

Algae are no doubt the material of the future. Their production is a new branch of agriculture. They are already being used in the food, cosmetic and feed industries. They are also used in the production of fertilizers, biofuels or in wastewater treatment processes, among others. Therefore, the topic of this research is in accordance with current trends [8]. In the available literature, there are few publications on the mineral content of spirulina. Our study showed that the most abundant minerals in spirulina supplements were K and P, and the least abundant was Se. A similar trend was notified by Rutar et al. [25] in spirulina supplements accessible on the Slovenian market. Shaban et al. [30] observed a lower content of P but higher content of K. Furthermore, the authors reported that the supplements they studied were a richer source of Fe, but contained less Na, Ca and Mg than the supplements we studied. Tolpeznikaite et al. [31] determined much higher concentrations of Na, Mg, K and Fe. Higher levels of Se, Fe, Mg and Zn, but lower amounts of Na and K, compared to those in the spirulina supplements analyzed in this study, were reported by Moheimani et al. [32]. Tokusoglu et al. [33] compared the mineral content of three algae: Spirulina, *Chlorella* and *Isochrisis*. Two of the spirulina samples they tested had a lower Na content, and all had lower contents of P and Zn, similar amounts of K, Ca and Mg, and a higher content of Fe and Se. *Isochrisis* samples, on the other hand, contained less K, Na, P and Zn and more Ca, Mg, Fe and Se. Microalgae of the genus *Chlorella* were characterized by lower contents of K, Ca and Zn, similar levels of Na and Mg and higher contents of P and Se. Given the fact that selenium can be harmful in excessive amounts, spirulina is a safer choice in this respect. Bito et al. [34] indicated that *Chlorella* has a lower content of Na and Zn and a higher Fe content. Salleh et al. [35] in turn examined the mineral content of various algal species, though *Arthrospira platensis* was not among them. A comparison of our own findings with those of other authors regarding the mineral content of spirulina is shown in Table 7.

**Table 7.** Element content of spirulina supplements—comparison of our research with data from the literature.

| | Elements Content [mg/kg] | | | | | | | | |
|---|---|---|---|---|---|---|---|---|---|
| Rerefences | Ca | Fe | Mg | K | Na | Sr | Zn | P | Se |
| [Our study mean] | 8554.46 | 664.8 | 3726.3 | 14,274 | 13,439.4 | 42.83 | 45.96 | 15,149.59 | 0.31 |
| [25] | 460–63,500 | 370–3480 | | 5830–26,900 | | 4.39–478 | 2.3–52.7 | 5060–14,700 | 0.0–2.7 |
| [30] | 1400–3500 | 230–1200 | 1600–3100 | 13,900–21,700 | 5600–7000 | | | 1500–3200 | |
| [31] | | | | | | | | | |
| Spirulina 1 | 71,700 | 1370 | 15,500 | 110,000 | 136,000 | 60–346 | | | |
| Spirulina 2 | 29,100 | 4730 | 51,000 | 950,000 | 458,000 | | | | 4.0 |
| [32] | 1.57 | 8998 | 4311 | 7680 | 1637 | | 1673 | | 1.31 |
| [33] | | | | | | | | | |
| Spirulina 1 | 8830 | 901 | 3986 | 13,269 | 18,973 | | 24.5 | 7034 | 36.8 |
| Spirulina 2 | 8930 | 924 | 3683 | 15,040 | 9886 | | 25.7 | 7460 | 1.3 |
| Spirulina 3 | 7030 | 1036 | 3997 | 14,080 | 9023 | | 30.1 | 8027 | 1.1 |
| Chlorella | 593.7 | 259 | 344.3 | 49.92 | 1346.4 | | 1.19 | 1761.5 | 0.07 |
| Isochrisis | 1081 | 228.4 | 688.6 | 1193.2 | 1109.2 | | 2.74 | 1252.4 | 1.02 |

Both in this study and in the research by other authors there are marked differences in mineral content between samples of individual supplements. These differences may be attributable to a multitude of reasons, including different growing conditions (under natural conditions or in media under controlled conditions) and processing methods of the microalgae. The method used to determine the mineral content, including the equipment used and the accuracy of the measurement, can also influence the results. Nevertheless,

we have not found any studies that would address the effect of supplement form or cultivation method on the mineral content of spirulina supplements. The only paper of this type was authored by our team and described the fluoride content of spirulina supplements. In this study, we found that the fluoride content in supplements in tablet form was significantly higher. No statistically significant differences in the fluoride content were observed depending on the method of cultivation [14]. Publications on various plant raw materials confirm that the method of their cultivation is one of the factors affecting the content of elements [36,37].

Our findings show that both supplement form and cultivation method affect the mineral content of spirulina. Supplements in powder form had a significantly higher content of important elements, such as Fe, Mg and K, while at the same time containing significantly less Na. This is an important observation, given that the consumption of Na, whose main dietary source is table salt, is often alarmingly high. Consequently, supplements in powder form appear to be a far better choice. In terms of the cultivation method, organic spirulina supplements turned out to be a richer dietary source of Fe and K. In turn, conventionally grown supplements had higher contents of Ca, P and Sr. It should also be noted that organic spirulina supplements were only available in tablet form. Thus, in the comparison of the mineral content of spirulina tablets originating from organic vs. conventional cultivation, it was found that organic spirulina tablets contained more Fe, Mg and K.

Manufacturers differ in their recommendations regarding the intake of spirulina supplements. The recommended daily dosage of spirulina usually ranges from 2 to 3 g (4–6 servings per day of 500 mg each), and can go up to 10 g, with a maximum daily limit for consumption of 30 g [38].

When writing about spirulina as a source of minerals, it is worth highlighting their importance in the diet. In the Discussion, the authors will refer to the recommended intake of 2–3 g, as these values are the most common in the literature. Therefore, when referring to the percentage of the daily requirement for a given mineral, the first value refers to the intake of 2 g of spirulina and the second value to the intake of 3 g of spirulina per day.

Ca serves as a building block of not only bones and teeth, but also connective tissue. It is also involved in blood coagulation, conduction of nerve impulses and regulation of certain hormones [39]. Its recommended daily intake is between 1.0 and 1.20 mg/day for men and women. Spirulina supplements cover approximately 2% of the requirement for this mineral. P is one of the most important elements in the human body. It is responsible for the normal structure of bones, forming compounds with Ca, and is a component of ATP, nucleic acids and phospholipids [40]. The recommended daily intake of P is 700 mg/day for both men and women. The studied supplements cover on average from 4% to 6% of the daily requirement for this mineral. Mg is very important for the normal functioning of the human body. It is involved in the biochemical processes that regulate the functioning of the circulatory, endocrine, digestive and bone and joint systems. It also plays a role in nerve conduction [41]. The requirement for Mg is 420 mg/day for men and 320 mg/day for women. The studied supplements cover between 1.8% and 2.7% of the requirement for men and 2.3–3.5% for women (when taking 2 g or 3 g of spirulina per day, respectively). A diet containing the optimal amounts of K can have a positive effect on lowering blood pressure [42]. Excess K is excreted by the kidneys, and hence its high levels in people with kidney disease are dangerous [43]. The studied supplements cover a small percentage of the K requirement, as little as 0.2–0.3%. Table salt is the primary source of sodium in the human diet. According to the WHO, one should consume no more than 2 g of Na per day, which is the equivalent of 5 g of salt (one teaspoon) [44]. Given that salt is now added to most products, e.g., bread and cold cuts, most people consume much higher amounts of sodium than recommended. Excessive amounts of Na in the diet cause hypertension and consequently increase the risk of cardiovascular disease [45]. The studied supplements covered less than 1% of the sodium requirement, but given its large amounts in foods, this is a good thing. Blood loss due to gastrointestinal bleeding and heavy menstruation

are the most common causes of Fe deficiency, leading to anemia. The main symptoms of Fe deficiency include chronic fatigue, hair loss and restless legs syndrome [46]. Fe is responsible, among other things, for transporting oxygen and electrons. It has been proven to affect brain development and function throughout life [47]. The Fe requirement for men is 10 mg/day and for women is 18 mg/day. The studied supplements cover as much as 13–20% of the requirement for men and 7–11% for women. Se has recently become increasingly popular as a mineral with anticancer and anti-inflammatory properties. It has been suggested that it will play a key role in cancer therapy and prevention in the future, but this opinion has yet to be confirmed by thorough research [48]. It is important to bear in mind that Se is involved in the regulation of numerous metabolic pathways and its action is influenced by various factors. Further, it is important to note that excessive Se intake can be toxic, hence caution is advised in its supplementation [49]. The recommended daily intake of selenium is 55 μg/day for men and women. When taken as recommended, the studied supplements cover on average 1.1–1.7% of the daily requirement, which is a small percentage and should be safe in the absence of other selenium-containing supplements. The main dietary source of Sr is water and certain foods [50]. The element plays a role in preventing the onset of osteoporosis and is also effective in treating tooth sensitivity as well as having a cardiostatic effect [51,52]. There are also emerging studies that link higher Sr concentrations in the body to a lower risk of type 2 diabetes [50]. Nevertheless, the recommended daily intake for Sr has not been determined.

The requirement for some minerals, such as Ca, Fe, Mg and K, can be difficult to meet from food alone. Spirulina can therefore be a valuable addition to the daily diet, helping achieve the required intake of micronutrients.

## 5. Conclusions

It has been shown that spirulina supplements can be a valuable addition to the daily intake of minerals, especially P, Fe, Mg and K, which are extremely important for healthy nutrition. Both the method of spirulina cultivation and the form of the supplement affected the mineral content. At the same time, no basis was obtained for a clear indication of which supplements are the richest source of a particular element. It seems that a good recommendation would be for supplement manufacturers to declare the content of key minerals on their labels, so that the consumer is aware of how much of a particular mineral he or she is consuming with the supplement.

**Supplementary Materials:** The following supporting information can be downloaded at: https://www.mdpi.com/xxx/s1, Table S1. Element content of tested samples (*n* = 33) of spirulina supplements-results of statistical analysis. Table S2. Element content in spirulina supplements depending on the form of the supplement. Table S3. Element content of spirulina supplements depending on cultivation method. Table S4. Element content of spirulina supplements in tablets depending on the of cultivation method.

**Author Contributions:** Conceptualization, K.J.-M.; methodology, P.K. and B.P.; formal analysis, K.S.-Ż. and K.J.; investigation, A.T.-M., P.K., A.L. and B.P.; resources, E.S. and B.D.; writing—original draft preparation, K.S. and B.D.; writing—review and editing, K.J.-M.; supervision, K.J.-M.; project administration, K.J.-M.; funding acquisition, E.S. All authors have read and agreed to the published version of the manuscript.

**Funding:** This research was funded by the Pomeranian Medical University in Szczecin, Poland (Project numer: WNoZ 330-01/S/2022).

**Institutional Review Board Statement:** Not applicable.

**Informed Consent Statement:** Not applicable.

**Data Availability Statement:** Not applicable.

**Conflicts of Interest:** The authors declare no conflict of interest.

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
