# Peer review of "Spirulina Supplements as a Source of Mineral Nutrients in the Daily Diet"

_applsci, doi:10.3390/app13021011_

Round 1

Reviewer 1 Report

Paper applsci-2126768

Title: Effect of cultivation method and supplement form on mineral content in spirulina

Review report

This paper is a research article that proposed the study of the influence of spirulina supplement format and cultivation on its mineral composition. The idea proposed is interesting by the results discussion is not sufficiently discussed – differences and correlations are merely presented without reasoned arguments and hypotheses. Article should be bonified before publication, specially the results and discussion section, so that conclusion may be taken from the study.

Specific comments

Abstract

Line 28: Verify the use of “elements” in a food perspective. Maybe change it by “micronutrient”.

I suggest the authors add in the abstract that “cultivation” method is classified in conventional and organic, as this the reader could wait for other cultivation parameters of spirulina. Also specify in parenthesis the supplement forms evaluated (powder, tablet)

Keywords: Always propose different keywords from title to increase article’s searchability.

Methods

Lines 170-171: Correct chemical formulas (subscripts).

Line 183: How did the authors check homoscedasticity of data? Were all datasets normal?

Results

Line 195: The two decimals are not relevant of values in the range of hundreds and thousands. Please revise the recommendations of decimals regarding the size of the number and the scientific significance of them.

Lines 195-196: Why were the standard deviations larger than the values for phosphorus and calcium? These values are not adequate as they represent far more than 50% of the average.

Table 4: Standard deviation values should be added to the table – and if standard deviations are too high, statistical analysis (regarding differences among averages) may not be valid. I suggest adding these tables (3, 4, 5, 6) in Supplementary data and then present data in a simplified form (only mean, standard deviation, and letters (to represent statistical difference)) in the manuscript, per form. Simplified tables make it easier for the reader to grab the relevant information from the data. Although statistical tables are important, they are difficult to interpret by most readers. Also, it is important to add full details on the tests used to determine significant difference.

Tables 7 and 8: I have many concerns with Table 7. Firstly, correlations should only be run between parameters for which authors have a hypothesis. Secondly, “correlations” of less than 0.60 are not correlated, thus should not be presented. I suggest the authors add in the text directly, and only the relevant correlations (and explained!!).

Lines 276-278: Please leave only correlations that are explained by hypothesis and strong discussion, otherwise, they should be removed.

Page 10: As the correlations are presented in Page 10, they are not sufficiently discussed because there are no explanations for each one of the correlations – they are only cited. Please remove or try to explain each correlation with grounded arguments.

Table 9: This table is interesting because groups much information from the literature. However, authors should provide a deepen discussion on why those differences are observed. Try to add reasoned arguments to support your hypothesis.

Line 352: There is space between the number and the “g”.

Author Response

Responses to Reviewer 1

Thank you for any comments that will help improve the value of our manuscript.
In accordance with the Reviewer's suggestions, the manuscript was restructured and supplemented.

Abstract

Line 28: Verify the use of “elements” in a food perspective. Maybe change it by “micronutrient”. It is not possible to change the term "elements" to "micronutrient" because the content of not only micronutrients was determined. A frequently used term, including in MDPI publishing journals, is "mineral nutrients" - we use this term in the title.

I suggest the authors add in the abstract that “cultivation” method is classified in conventional and organic, as this the reader could wait for other cultivation parameters of spirulina. Also specify in parenthesis the supplement forms evaluated (powder, tablet) done

Keywords: Always propose different keywords from title to increase article’s searchability. done

Methods

Lines 170-171: Correct chemical formulas (subscripts). done

Line 183: How did the authors check homoscedasticity of data? Were all datasets normal? This is described in the Methodology. The Shapiro-Wilk test was used to assess whether the distribution of the data follows a normal distribution: “Distributions of values for individual parameters were analyzed using the Shapiro-Wilk test. Since the distribution of continuous variables deviated from normal, the Test U Manna-Whitneya test was used to evaluate the differences between the studied parameters”.

Results

Line 195: The two decimals are not relevant of values in the range of hundreds and thousands. Please revise the recommendations of decimals regarding the size of the number and the scientific significance of them. The Reviewer is right - with such large values, giving data to the second decimal place is unreasonable – corrected

Lines 195-196: Why were the standard deviations larger than the values for phosphorus and calcium? These values are not adequate as they represent far more than 50% of the average. In many cases, the standard deviations are very large. This is due to the fact that the tested samples of supplements were characterized by very different contents of elements.

Table 4: Standard deviation values should be added to the table – and if standard deviations are too high, statistical analysis (regarding differences among averages) may not be valid. I suggest adding these tables (3, 4, 5, 6) in Supplementary data and then present data in a simplified form (only mean, standard deviation, and letters (to represent statistical difference)) in the manuscript, per form. Simplified tables make it easier for the reader to grab the relevant information from the data. Although statistical tables are important, they are difficult to interpret by most readers. Also, it is important to add full details on the tests used to determine significant difference. The original tables (Tab. 3-6) have been moved to the supplements, and abbreviated tables have been included in the text

Tables 7 and 8: I have many concerns with Table 7. Firstly, correlations should only be run between parameters for which authors have a hypothesis. Secondly, “correlations” of less than 0.60 are not correlated, thus should not be presented. I suggest the authors add in the text directly, and only the relevant correlations (and explained!!).

Lines 276-278: Please leave only correlations that are explained by hypothesis and strong discussion, otherwise, they should be removed.

Page 10: As the correlations are presented in Page 10, they are not sufficiently discussed because there are no explanations for each one of the correlations – they are only cited. Please remove or try to explain each correlation with grounded arguments. Correlations between elements were not the main purpose of the paper. In addition, they add relatively little to the manuscript and it is difficult to provide a discussion of the data obtained, so the authors decided to remove this section.

Table 9: This table is interesting because groups much information from the literature. However, authors should provide a deepen discussion on why those differences are observed. Try to add reasoned arguments to support your hypothesis. The probable reasons for the differences in results are presented in the Discussion section. I cite that section: “Both in this study and in the research by other authors, there are marked differences in mineral content between samples of individual supplements. These differences may be attributable to a multitude of reasons, including different growing conditions (under natural conditions or in media under controlled conditions) and processing methods of the microalgae. The method used to determine the mineral content, including the equipment used and the accuracy of the measurement, can also influence the results. Nevertheless, we have not found any papers that would address the effect of supplement form or cultivation method on the mineral content of spirulina supplements”. Literature has also been added.

Line 352: There is space between the number and the “g”. There must be a space between the number and the "g". The whole text has been checked and errors corrected

Reviewer 2 Report

In the manuscript entitled “Effect of cultivation method and supplement form on mineral content in spirulina”, the authors investigated the effect of supplement form and cultivation method on the mineral content of spirulina. Overall, this manuscript is demonstrating an important and promising research direction, however part of this manuscript seems to be enhanced in much clearer discussion. Herein, it is suggested that the manuscript could be accepted for publication in Applied Sciences unless major corrections has been conducted according to the recommended points.

Required corrections:

1) Title:

The title is unattractive. It does not reflect the greatness of the work. I suggest they use more interesting terms.

Besides, spirulina should be expressed in italics.

2) Abstract:

The current research gap about this study should be mentioned at the beginning.

Besides, it is suggested that the authors should emphasize and point out the numerical conclusions, not only the importance and significance of this study. Please balance this situation.

3) Keywords:

Microalgae technology should be added.

4) Introduction:

This section needs to be modified and separated into 3~4 paragraphs. The first paragraphs should give a brief introduction and summarization of microalgae, such as the advantages and current innovations in microalgae technology, including the cultivation method, dietary supplement, renewable biofuel generation, and so on. It is suggested that some recent literature related to could be referred to, such as (a) (b) and (c): (a) Journal of Cleaner Production, 2022, 355, 131768; (b) Bioengineering, 2022, 9(11), 637; (c) Biochemical Engineering Journal, 2022, 179, 108330.

5) Results:

It is suggested that the authors should list the advantages and comparison of this study through comparing the current studies.

6) Discussion:

Future perspective should be added in this section.

7) Text must grammar improves and in some cases it is very weak.

8) There are many format errors which suggested the authors need to be modified carefully throughout the whole manuscript.

Author Response

Responses to Reviewer 2

Thank you for any comments that will help improve the value of our manuscript.
In accordance with the Reviewer's suggestions, the manuscript was restructured and supplemented.
As for the language proofreading of the manuscript, it is necessary first to accept all changes by Reviewers and the Editorial Board

1) Title:

The title is unattractive. It does not reflect the greatness of the work. I suggest they use more interesting terms. Title has been changed to: Spirulina supplements as a source of mineral nutrients in the daily diet

Besides, spirulina should be expressed in italics. Latin is used for genus and species names; spirulina is not a genus or species name, the Latin species name is Arthrospira - in the text it is written in Latin

2) Abstract:

The current research gap about this study should be mentioned at the beginning. added

Besides, it is suggested that the authors should emphasize and point out the numerical conclusions, not only the importance and significance of this study. Please balance this situation. Added numerical values in the abstract

3) Keywords:

Microalgae technology should be added. added

4) Introduction:

This section needs to be modified and separated into 3~4 paragraphs. The first paragraphs should give a brief introduction and summarization of microalgae, such as the advantages and current innovations in microalgae technology, including the cultivation method, dietary supplement, renewable biofuel generation, and so on. It is suggested that some recent literature related to could be referred to, such as (a) (b) and (c): (a) Journal of Cleaner Production, 2022, 355, 131768; (b) Bioengineering, 2022, 9(11), 637; (c) Biochemical Engineering Journal, 2022, 179, 108330. text has been completed

5) Results:

It is suggested that the authors should list the advantages and comparison of this study through comparing the current studies. A comparison of our study results with those of other authors can be found in the Discussion section

6) Discussion:

Future perspective should be added in this section. Added

Reviewer 3 Report

The manuscript entitled “Effect of cultivation method and supplement form on the mineral content in spirulina” is an interesting idea, however, it needs substantial improvement. 

I have serious concerns about the title, the title should be informative and must ba attractive rather than a simple statement.

The abstract section is written poorly: authors have given many unnecessary information at the start of the abstract. This information must be deleted, and authors must add experiment details (treatments e.g., cultivation) along with the numerical values.

The introduction section needs information about the effect of cultivation (conventional vs organic growing) on the mineral composition of Spirulina, besides the author should also add the hypothesis of the study before the objectives in the introduction section.

The authors have obtained 33 spirulina samples from different countries of origin, I suggest the authors to add the names of countries in Table 1. If possible please also add the composition of all 33 tested samples.

The statistical analysis used by the authors gives no clear difference, I suggest the authors to differentiate the difference in conventional and organic cultivation using appropriate difference text (LSD, Tukey etc......).

The results and discussion sections are well written, however, authors must have a strong conclusion/takeaway message for the readers.

There are some mistakes in English grammar, syntax, and punctuation that distract the readers, therefore, please carefully review your manuscript prior to re-submission.

Author Response

Responses to Reviewer 3

Thank you for any comments that will help improve the value of our manuscript.
In accordance with the Reviewer's suggestions, the manuscript was restructured and supplemented.
As for the language proofreading of the manuscript, it is necessary first to accept all changes by Reviewers and the Editorial Board

I have serious concerns about the title, the title should be informative and must ba attractive rather than a simple statement. Title has been changed to: Spirulina supplements as a source of mineral nutrients in the daily diet

The abstract section is written poorly: authors have given many unnecessary information at the start of the abstract. This information must be deleted, and authors must add experiment details (treatments e.g., cultivation) along with the numerical values. Numerical values added. We are unable to provide details of the cultivation, as the subject of the study was purchased spirulina supplements from stores. The abstract has been shortened.

The introduction section needs information about the effect of cultivation (conventional vs organic growing) on the mineral composition of Spirulina, besides the author should also add the hypothesis of the study before the objectives in the introduction section. Information on major producers has been added and cultivation methods have been mentioned

The authors have obtained 33 spirulina samples from different countries of origin, I suggest the authors to add the names of countries in Table 1. The name of the country in which the company supplying the supplements is registered has been added.

If possible please also add the composition of all 33 tested samples. The tables show minimum, maximum, mean and median values, so it seems unnecessary to give all the results.

The statistical analysis used by the authors gives no clear difference, I suggest the authors to differentiate the difference in conventional and organic cultivation using appropriate difference text (LSD, Tukey etc......). The Tukey test and LSD are suitable for analyzing results that follow a normal distribution. The data we obtained does not follow a normal distribution so the Mann-Whitney U test, dedicated to analyzing results that deviate from the normal distribution, was used. 

The results and discussion sections are well written, however, authors must have a strong conclusion/takeaway message for the readers. The Conclusion section has been added

Reviewer 4 Report

Dear Authors,

I reviewed your article titled in (Effect of cultivation method and supplement form on mineral content in spirulina). Overall, the data presented here is valuable to those working in this field demonstrates the effectiveness of a relatively simple intervention that could be applied a wider scale especially in the field of supplemental human nutrient. A thorough revision of the English grammar, sentence structure and deep editing of some parts needs to be undertaken before this is at all ready for publication. There are some other points that should be addressed in the individual sections, which I have specified below and in attached pdf file:

Title, Abstract, and introduction: 

1) Line 25-27: too long. Please revised to be more concluded and add this part in introduction.

2) lines 42-43: add conclusion result.

3) Lines 44: use different words than title and arrange alphabetically.

4) Lines 47-48: As i suggest, repeated information. Please revised.

5) Line 89: Provides the safety of this product from FDA organization.

Materials and methods:

This section of results is overall well written. Some comments are needed which mention here:

1) line 129: model, country

Results and Discussion

This section (results) is overall well written. However, the discussion is needed deep revision such as:

1- use abbreviation of the elements for the whole section

2- The name of author should be followed by the number to can follow with you.

3- There is a deep leakage of references which start in line 356. You should add references after all information that you present.

All these comments are presented in attached pdf file.   

Conclusion

this section is missed. Create a chart to conclude your results.

Author Response

Responses to Reviewer 4

Thank you for any comments that will help improve the value of our manuscript.
In accordance with the Reviewer's suggestions, the manuscript was restructured and supplemented.

Title, Abstract, and introduction: 

1) Line 25-27: too long. Please revised to be more concluded and add this part in introduction. corrected

2) lines 42-43: add conclusion result.

3) Lines 44: use different words than title and arrange alphabetically. done

4) Lines 47-48: As i suggest, repeated information. Please revised. corrected

5) Line 89: Provides the safety of this product from FDA organization. Correct, nevertheless, many food products whose safety is assured by the FDA are the subject of scientific research; this is not excluded

Materials and methods:

This section of results is overall well written. Some comments are needed which mention here:

  • line 129: model, country – added

Results and Discussion

This section (results) is overall well written. However, the discussion is needed deep revision such as:

1- use abbreviation of the elements for the whole section - done

2- The name of author should be followed by the number to can follow with you. done

3- There is a deep leakage of references which start in line 356. You should add references after all information that you present.  corrected

Conclusion

this section is missed. Create a chart to conclude your results. added

Round 2

Reviewer 1 Report

All required changes were made.

Author Response

Thank you very much for all your suggestions, which helped us to improve the quality of the manuscript.

Reviewer 2 Report

The revised manuscript is clear and significantly improved. Acceptance is recommended unless some errors have been corrected:

1) Some of the references format and infromation were incomplete and need to be supplemented by authors, such as the journal of references are missing (Line 583-584).

Author Response

Thank you very much for all your suggestions, which helped us to improve the quality of the manuscript

Reviewer 3 Report

The authors have substantially improved the MS as per suggestion therefore, now it can be accepted for publication. 

Author Response

(The authors gave the same response as above.)

Reviewer 4 Report

Dear authors,

thank you for your improvement for your article. However, there are still some comments that you need to modified. I highlighted the comments in attached pdf file that you upload after your first review. Also, I suggest to revised your article by a native speaker or by MDPI office center. 

All the best

Author Response

Thank you for all your comments, which helped to improve the quality of our manuscript. All of the Reviewer's comments in Round 2 have been taken into consideration and included in the manuscript.